



# An Extended Radar Relative Calibration Adjustment (eRCA) Technique for Higher Frequency Radars and RHI Scans.

Alexis Hunzinger[1], Joseph C. Hardin[1], Nitin Bharadwaj[1], Adam Varble[1], and Alyssa Matthews[1]

[1]Pacific Northwest National Laboratory, Richland, WA, USA

**Correspondence:** Alexis Hunzinger (a.e.hunzinger@gmail.com)

**Abstract.** This study extends the relative calibration adjustment technique for calibration of weather radars to higher frequency radars, as well as range-height indicator scans. The calibration of weather radars represents one of the most dominant sources of error for their use in a variety of fields including quantitative precipitation estimation and model comparisons. While most weather radars are routinely calibrated, the frequency of calibration is often less than required resulting in miscalibrated time

periods. While full absolute calibration techniques often require the radar to be taken offline for a period of time, there have been online calibration techniques discussed in the literature. The relative calibration adjustment (RCA) technique uses the statistics of the ground clutter surrounding the radar as a monitoring source for the stability of calibration but has only been demonstrated to work at S- and C-Band for plan-position indicator scans at a constant elevation.

In this work the RCA technique is modified to work with higher frequency radars, including Ka-band cloud radars. At higher

frequencies the properties of clutter can be much more variable. This work introduces an extended clutter selection procedure that incorporates the temporal stability of clutter and helps to improve the operational stability of RCA for relatively higher frequency radars. The technique is also extended to utilize range-height scans from radars where the elevation is varied rather than the azimuth. These types of scans are often utilized with research radars to examine the vertical structure of clouds.

The newly extended technique (eRCA) is applied to four DOE-ARM weather radars ranging in frequency from C- to Ka-

band. Cross comparisons of three co-located radars with frequencies C, X, and Ka at the ARM CACTI site show the technique can determine changes in calibration. Using an X-band radar at the ARM Eastern North Atlantic site, we show how the technique can be modified to be more resilient to clutter fields that show an increased variability, in this case due to sea clutter. The results show that this technique is promising for a-posteriori data calibration and monitoring.

## 1 Introduction

The Department of Energy's Atmospheric Radiation Measurement (ARM) program deploys weather radars around the world to observe a variety of weather regimes. These radars operate in remote regions with limited on-site monitoring. Without the ability to conduct routine calibrations, weather radars can experience a degradation in calibration accuracy over time. To





mitigate this, a robust technique to monitor and correct calibration drifts is needed, as well as a tool to monitor for possible periods of reduced performance. Additionally, for historical campaign data where retroactive calibrations cannot be performed *a-posteriori*, there is a need to flag periods of miscalibration. Previous and existing calibration techniques rely on the detection or measurement of a known target (i.e. metallic spherical targets (Willis et al., 1964; Glover et al., 1966; Stratmann et al., 1971),

or the presence of precipitation (i.e. evaluation of the drop size distribution (Marks et al., 1993; Gage et al., 2000). While these methods are effective, they require the radar to be offline or depend on weather conditions and account only for a brief period of time. A promising online method, based loosely on a technique in Rinehart (1978), is presented by Silberstein et al. (2008) as the relative calibration adjustment (RCA) technique. This method uses ground clutter, which allows for continuous monitoring of radar health since ground clutter is a persistent radar signature. While this technique has been successfully applied to S- and

C-band weather radars performing traditional plan-position-indicator (PPI) scans, there are challenges extending it to cloud radar frequencies and alternate scan types.

The RCA technique utilizes changes in the probability distribution of ground clutter reflectivity at the lowest elevation to identify changes in the radar system. This technique allows for tracking of radar calibration without having to take a radar offline, which is important for research radars operating during field campaigns when observation periods are limited and

historical datasets for which the capability to retroactively calibrate is limited. While the reflectivity of individual clutter targets can change significantly depending on environmental conditions (rain-free, light or heavy precipitation, anomalous propagation), the 95th percentile of the probability distribution of ground clutter reflectivity remains stable in the absence of engineering changes, system failures (i.e. failure of radar components), or pointing angle errors (Silberstein et al., 2008). Based on this, RCA (Silberstein et al., 2008) can track changes in calibration regardless of environmental conditions.

Previously, Silberstein et al. (2008) performed the RCA technique on the Kwajalein polarimetric S-band weather radar (KPOL) in an oceanic tropical environment with success, while Wolff et al. (2015) successfully applied the method to the NASA polarimetric S-band ground-based radar (NPOL) for three different Global Precipitation Measurement (GPM) mission field campaigns across the continental United States. Louf et al. (2019) applied the RCA technique to C-band radar data collected at Darwin, Australia and validated the use of the 95th percentile of ground clutter reflectivity for monitoring the radar

calibration over a longer time period.

The basis for the RCA technique relies on the assumption that any variation in ground clutter reflectivity is caused by a change in radar calibration. Persistent ground clutter echoes at low elevation angles generally come from features in the landscape surrounding the radar. Sources of ground clutter can include trees, buildings, towers, or terrain. Often the radar signatures for these features are high reflectivity ($Z$), zero velocity ($\nu$; stationary objects, with the exception of wind-blown

trees), and low correlation coefficient ($\rho_{hv}$; non-meteorological). While RCA has been applied to many different radars at S- and C-band, the properties of clutter are very dependent on radar frequency, as shown in Skolnik (2000).

The RCA technique in its current form is not compatible with all ARM radars and their deployments. In this paper, the RCA technique is adapted for and applied to three radar frequencies, C-, X-, and Ka-band, operating at two ARM sites. This extended RCA (eRCA) technique aims to support a wider range of radar frequencies and scan types by modifying the clutter selection





procedure. The standard scan strategies of these research radars include plan-position indicator (PPI) and range-height indicator (RHI) scans. The RCA technique, as originally proposed does not support the use of RHI scans. This paper demonstrates the development and validation of the RCA technique for RHIs using data collected with a scanning C-band radar deployed during a DOE ARM field campaign in the Sierras de Córdoba mountain range of Argentina. The RCA technique is then expanded and

evaluated for use on higher frequency (X- and Ka-band) radars, the first known application of its kind. The application to RHI scans is of special interest as ARM radars do not always conduct PPI scans for field campaigns.

Section 2 describes the datasets used in this study. Section 3 outlines the clutter selection procedure for developing a clutter map, followed by the introduction of an extended clutter selection procedure that helps to improve the operational stability of eRCA for higher frequency radars, ending with requirements for baseline selection and eRCA calculation. Section 4 presents

usage of the extended clutter map selection procedure, a comparison of calibration adjustments derived from the same radar using PPIs and RHIs, and a comparison of RHI-derived eRCA values from C-, X-, and Ka-band radars located at the same site, before discussing limitations of the algorithm. Section 5 summarizes conclusions and suggests future applications of the new methods created in this study.

## 2 Radar Datasets

The United States Department of Energy's Atmospheric Radiation Measurement (ARM) user facility runs multiple fixed and mobile atmospheric observing facilities spread through the world. A wide variety of weather radars are routinely deployed at these sites as part of long term measurements, as well as shorter field campaigns. While the choice of radar configuration at each site depends on the science questions of that site, it is common to have scanning precipitation and cloud radars deployed at the majority of the sites. This work will focus on two sites in particular to highlight improvements made to the RCA algorithm; the

Eastern North Atlantic (ENA) site and Argentine site during the Cloud, Aerosol, and Complex Terrain Interactions (CACTI) field campaign. Both of these ARM sites are located in regions with prominent and persistent ground clutter signatures, making them ideal for eRCA technique testing and evaluation.

The RCA technique relies on ground clutter statistics obtained from radar reflectivity. As such, all data used is prior to any clutter filtering applied by the radar processing. All scans are pulled from regular operations of the radar.

### 2.1 Eastern North Atlantic ARM site

The ENA ARM measurement site has been located on Graciosa Island in the Azores archipelago west of Portugal since 2013. Observations from this site provide a rare long-term dataset of marine stratocumulus clouds and their interactions with atmospheric aerosols. A suite of ground-based in-situ and remote sensing instruments are used to characterize meteorological conditions, aerosols, surface state, clouds, precipitation, and radiation.

The site has hosted a high-powered, dual-polarization X-Band Scanning Precipitation Radar (XSAPR2) to provide spatial context and measure drizzle and rain rates since 2017. The details of this radar can be found in Table 1. The XSAPR2 is



positioned on the northern coast of the island. Terrain on the island creates blockage to the south of the radar. For this reason, only the northern sector is scanned to avoid beam blockage.

We utilized almost 3 months of data collected with XSAPR2 in 2018 (13 January - 31 March) to evaluate the eRCA technique. Immediately prior to this time period, the radar was visited by engineering staff and maintenance was performed just
before the start of the ACE-ENA field campaign (Wang et al., 2019). The data consists entirely of PPI sector volumes taken at elevations from 0 to 5 degrees. The XSAPR2 sector PPIs reveal a ground clutter signature in close proximity to the radar attributed to terrain. Additionally there is a large contribution from sea spray that changed and shifted spatially over time, often with the predominant wind conditions. Figure 1 is a sample PPI from XSAPR2 and an example of the ground clutter contribution from sea spray to the north of the radar within 20 km. This sea spray was flagged by the original RCA technique
as ground clutter, and due to its shifting properties, causes the technique to fail. This evolving and intermittent clutter signature prompted the development of the composite clutter map method, described in Section 3.2.

## 2.2    CACTI: Sierras de Córdoba ARM site

The CACTI field campaign (Varble et al., 2019) took place in the Sierras de Córdoba mountain range of central Argentina from 15 October 2018 - 30 April 2019. The campaign studied convective cloud life cycle and organization in a region of the
world where orographic forcing along the Sierras de Córdoba mountain range enables repeated observations of deep convective initiation and upscale growth. The ARM site at this location (COR) was equipped with a variety of ground-based instrumentation similar to the ENA site, but was anchored by a fleet of radars. A total of four radars operated during the campaign, one vertically-pointing and three scanning. The radars operating at 3 different frequencies allow for the observation of crucial steps in the cloud life cycle, particularly for convective clouds and systems. Ka-band vertically-pointing and scanning radars capture
cloud evolution and precipitation formation. X-band scanning radar observes precipitation as the clouds mature and grow into a larger system. C-band radar surveys the spatial distribution of convective cells and mesoscale systems, as well as the vertical extent and structure of deep convective clouds and systems and associated precipitation. During the CACTI field campaign there was a mix of shallow and deep convection in the field of view of the radar. The three scanning radars offer an opportunity to evaluate the performance of the eRCA technique as applied to different frequencies and scan strategies.

### 25    2.2.1    CSAPR2

The C-band Scanning ARM Precipitation Radar (CSAPR2) is a dual-polarization Doppler weather radar that operates in a simultaneous transmit and receive mode, splitting the transmit signal so the power is transmitted in both horizontal and vertical polarizations at the same time. The CSAPR2 operates at a frequency of 5.7 GHz. Other specifications of CSAPR2 and the other radars used in this study are found in Table 1.

The CSAPR2 scan strategy includes hemispherical RHIs at six azimuths (0°- 180°, 30°- 210°, 60°- 240°, 90°- 270°, 120°- 300°, 150°- 330°). Each of these azimuths is considered in the development of the RHI clutter map. The lowest sweep (0.5 degree) of CSAPR2 PPI scans and the lowest 5 degrees of CSAPR2 RHI elevations are used for eRCA testing and execution.





The combination of RHI and PPI scanning provides a dataset for comparison of eRCA applied to the two scan types. Data collected by CSAPR2 between 8 November 2018 and 3 March 2019 are used in this study. Figure 2 is a PPI from CSAPR2, showing a time with no precipitation in the region and ground clutter signatures visible. In particular, the Sierras de Córdoba lie just west of the radar and produce a robust clutter signature.

### 2.2.2  Ka/X SACR

The Scanning ARM Cloud Radars (SACR) are dual-frequency, dual-polarization Doppler radars mounted on a common scanning pedestal. The SACR deployed at COR during CACTI includes an X- and a Ka-band radar operating in tandem. The X-band Scanning ARM Cloud Radar (XSACR) operates at a frequency of 9.71 GHz and the Ka-band Scanning ARM Cloud Radar (KaSACR) operates at a frequency of 35.3 GHz (see Table 1 for other radar specifications). The SACR primarily collects

hemispherical RHI scans at the same chosen azimuths as CSAPR2 to characterize the vertical structure of clouds and precipitation in the Sierras de Córdoba range. To compensate for mechanical issues with CSAPR2 during the last 2 months of the campaign, the SACR scan strategy included PPI scans in addition to the typical hemispherical RHI scans during this period. This study utilizes XSACR and KaSACR RHI scans from 15 October 2018 to 30 April 2019 to develop, test, and evaluate the eRCA technique for RHI scans and higher frequency radars.

## 3  Methodology

The RCA technique relies on the stability of the radar returns due to ground clutter. It was originally developed in Silberstein et al. (2008), and in this section we will cover the original implementation before introducing extensions to the technique to improve stability and adaptation to ARM radars.

The simplified form of the radar equation for meteorological targets (Bringi and Chandrasekar, 2001) calculates the equiva-

lent radar reflectivity factor, $z$ (mm$^6$ m$^{-3}$, henceforth referred to as reflectivity) as,

$$z = \frac{P_r r^2}{C} \qquad (1)$$

where $P_r$ (mW) is the returned power by the target, $r$ (km) is the range, and $C$ is the radar constant, which includes all other terms related to the radar, such as transmitted power, wavelength, pulse length, beam width, and antenna gain. These "constant" quantities are known for a particular radar, but can vary over time as a result of maintenance or degradation of radar hardware.

Working on a decibel scale, $z$ is written in dBZ ($10\log_{10} z = Z$ dBZ) to yield:

$$Z(\text{dBZ}) = 10\log P_r + 20\log r - 10\log C \qquad (2)$$

$Z$ is the sum of reflectivity contributed from precipitation ($Z_p$) and from ground clutter ($Z_c$). In the absence of scatterers in the atmosphere, the reflectivity is a function of the strength of the ground clutter. Ground clutter is characterized as persistent





echoes at fixed locations (i.e. mountains, buildings, trees), meaning further simplification of the radar equation may be done as $P_r$ and $r$ are considered constant for stationary targets with constant scattering properties. Ground clutter is assumed to not change significantly and as such, changes in ground clutter power can be related to changes in the radar constant by

$$\Delta Z_c = \Delta(10\mathrm{log}C) \tag{3}$$

After this simplification, any change in clutter reflectivity is attributable to changes in the radar system. By tracking the changes in ground clutter in a statistical sense, these changes can be directly tied to changes in radar configuration or calibration. Silberstein et al. (2008) demonstrated that a change in the calibration of a radar will manifest as an equivalent change in the 95th percentile of clutter power after masking. This will shift the probability density function (PDF) of the reflectivity by $\Delta(10logC)$ dB. This is the primary assumption of the RCA technique, after which a statistical analysis of ground clutter

reflectivity is used to monitor the radar calibration.

The following sections will detail the implementation of RCA, but generally the technique can be broken down into two stages. The first is generation of a clutter map to isolate known clutter points. This forms a baseline from which all changes are taken in a relative sense. The second step is to track the clutter over time and compare it to the baseline time period.

### 3.1   Clutter map development

The RCA technique utilizes the change in the probability distribution of ground clutter reflectivity compared to a baseline clutter reflectivity distribution to identify changes in the radar system and ultimately make an adjustment in the calibration. The adjustment is based on the 95[th] percentile clutter area reflectivity, defined in Silberstein et al. (2008). A clutter map is developed to determine the statistical distribution of clutter area reflectivity and only utilize gate locations that are associated with stable clutter signatures. We select a single precipitation-free day and set an appropriate reflectivity threshold for ground

clutter detection. This threshold is dependent on both the radar frequency and location, since the threshold must be greater than the bulk of the characteristic precipitation of the region. The following steps, originally outlined in Wolff et al. (2015) for S-band radar PPIs, are taken to develop the RCA clutter map.

1. Select all PPIs in a day with no precipitation within 5 km of the radar (see Table 2 for modification of range restrictions based on radar frequency and location).

2. Create a fixed polar grid/array (FPG) with a resolution of 1 km $\times$ 1° to serve as a mask for constructing the clutter map.

3. Using the 1 km $\times$ 1° elements of the the FPG array, flag each PPI pixel (i.e., radar gate with a range and azimuth) from the lowest tilt that exceeds a specified threshold of reflectivity. A value of one indicates that at least one PPI pixel within the FPG element exceeds the threshold and a value of zero indicates that no PPI pixels within the FPG element exceed the threshold.





4. Save the FPG array and repeat for all PPIs in a day.

5. Calculate a percentage of occurrence for each FPG element by summing the saved FPG arrays and dividing by the total number of arrays used. This value may be called "percent on" (PCT_ON) and indicates how often an FPG element contains at least one PPI pixel that exceeds the reflectivity threshold.

6. The final clutter map is defined as the range and azimuth locations where PCT_ON $\geq$ 50%. All other points are not considered clutter points.

These steps are translated into a work flow in Figure 3, which may be used as a template for developing the accompanying code where the numbered portions match the steps listed above.

Figure 4 shows the clutter map for the CSAPR2 radar during the CACTI field campaign. The filled 1 km $\times$ 1° elements are hereafter referred to as clutter map points, while the observed reflectivity within the clutter map points are referred to as the clutter area reflectivity. Only elements where PCT_ON $\geq$ 50% are considered clutter map points.

## 3.2 Composite clutter map development

Wolff et al. (2015) found that using a single precipitation-free day to develop the clutter map produced a stable 95th percentile clutter area reflectivity. However in some locations there are transient signals that can present similar to ground clutter that are not associated with a true ground clutter feature. This can be caused by sea clutter as in the ENA case, or transient changes in the environment. In these cases a clutter map produced from a single day may not reflect the appropriate distribution of clutter area reflectivity and could affect the 95th percentile value and resulting RCA, leading to inaccurate calibration adjustments.

To capture the most stable clutter signature and create a clutter map that is used to yield the most stable 95th percentile of the clutter area reflectivity distribution over an observing period, it is beneficial to create a composite clutter map using several single-day clutter maps from the observing period. The composite clutter map method ensures only the *most* stable clutter signatures are included when calculating clutter area reflectivity statistics. The following steps are taken to modify the RCA algorithm to use a new "composite" clutter map.

1. Create daily clutter maps by following the steps outlined in Section 3.1 for every Nth precipitation-free day. The value of N can be changed to address variability in the environment. A good value for N was derived empirically in this study to be five days.

2. For each day's clutter map, assign a value of one to all FPG elements that are considered clutter points (PCT_ON $\geq$ 50%). Assign a value of zero to all FPG elements that are not considered clutter points (PCT_ON $<$ 50%).

3. Sum the FPG elements of all days' clutter maps and divide by the number of days used (N) to obtain the percentage occurrence of clutter points (CMAP_ON). Retain the FPG elements that have a percentage occurrence of $>$ 80% (where CMAP_ON $>$ 0.8). These elements are considered the composite clutter map points.





These steps are translated into a work flow in Figure 3, which may be used as a template for the clutter map code. Numbering corresponds to the steps listed in Sections 3.1 and 3.2. There is no restriction on the number of daily clutter maps used to create the composite, so steps 1-2 may be repeated as many times as necessary. This method is proven most useful at an oceanic site where sea clutter contributes to a fluctuating clutter signature during the observing period. Section 4.1 details the benefits of
the composite clutter map at the ENA site.

### 3.3   Clutter map extension to RHI

Applications of the RCA technique thus far have only included the use of PPI scans (Silberstein et al., 2008; Wolff et al., 2015; Louf et al., 2019) due to the increase in clutter at low elevations scans. Operational radars most commonly use PPIs to get a broad look at nearby weather. However research radars tend to vary scan types depending on the goals of a particular field
campaign, often collecting RHIs in addition to PPIs. In the case of some radars, only RHI scans are conducted, relying instead on supporting area radars to provide PPIs. Applying the RCA technique to RHIs involves following the same clutter map steps from Sections 3.1 and 3.2 with a few adjustments to accommodate the difference in how clutter is represented in RHIs. RHI scan strategies target fewer azimuths and a larger range of elevations. To incorporate more data points for potential ground clutter, the range of interest is extended further than the PPI range limit and the lowest $5°$ of elevation angles are used. Table 2
lists the ranges and reflectivity thresholds used for different ARM radars and scan types used in this study.

Figure 5 shows the clutter map for CSAPR2 at COR, limited to azimuths that contained clutter map points. As with the PPIs, the filled 1 km $\times$ $1°$ elements are the clutter map points and any observed reflectivity within the clutter map points is considered clutter area reflectivity. Ranges and thresholds may be adjusted to develop clutter maps for different radar frequencies and scan types.

### 3.4   Baseline development

After a clutter map is constructed, the next step is to determine the baseline value. The baseline is the benchmark from which all other RCA values are compared. The use of a baseline is what makes this a *relative* calibration of radar reflectivity, and not an *absolute* calibration. The baseline day is usually chosen as a clear day, preferably close to a time with a known calibration value to simplify further analysis.

Using the selected baseline day, we calculate the probability density function (PDF) and cumulative distribution function (CDF) of the clutter area reflectivity using a single day clutter map (original technique) or the composite clutter map (extended technique). The PDF and CDF show the distribution of the clutter area reflectivity, which may differ from day to day, but stabilizes at the 95th percentile of the CDF (Silberstein et al., 2008). The clutter area reflectivity value at the 95th percentile of the CDF using the selected baseline day is denoted as $dBZ95_{baseline}$. Similarly, the 95th percentile clutter area reflectivity from





all other days in the dataset are denoted as $dBZ95_{daily}$. The RCA value is then calculated as the difference between the baseline 95th percentile clutter area reflectivity and the daily 95th percentile clutter area reflectivity.

$$RCA = dBZ95_{baseline} - dBZ95_{daily} \tag{4}$$

An RCA value is calculated for each day in the dataset and is plotted as a time series to track to relative change in clutter
area reflectivity on a day-to-day basis. If the RCA value is less than zero, the daily 95th percentile clutter area is greater than the baseline and the radar is considered to be biased high. If the RCA value is greater than zero, the daily 95th percentile clutter area reflectivity less than the baseline and the radar is biased low. This method is used for both PPI and RHI scans.

The work flow for baseline and RCA calculation are illustrated in Figure 6. Baseline and RCA development use the same method for calculating the 95th percentile clutter area reflectivity (gray box on left), requiring a day of radar files and the
previously developed composite clutter map as input. The baseline must be calculated first in order to be used in the daily RCA calculations.

Figure 7 shows the PDF and CDF of ENA XSAPR2 clutter area reflectivity for hourly scan times during the selected baseline day, 25 January 2018. The hourly distributions are in close agreement because no precipitation occurred during this day. The daily median 95th percentile clutter area reflectivity ($dBZ95_{baseline}$) here converges at 55.4 dBZ. Days with precipitation may
have varying PDF and CDF shapes, but still converge by the 95th percentile.

One concern when using RCA is the level of initial calibration. If the initial calibration state of the radar is significantly far off (say 10+ dB), then it is possible that points are not identified as clutter that otherwise should be. This can be mitigated by choosing a time period for the baseline that has a calibration that is believed to be close to correct.

The RCA technique in its current form has some limitations. Since the technique's strength lies in the persistence of ground
clutter, the absence of ground clutter makes this technique weak or useless in regions where there is a lack of ground clutter. Though no minimum has been determined, the technique requires enough persistent ground clutter points to function properly over a long period of time.

## 4   Results and Discussion

The extended RCA technique developed in Section 3 was applied to four radar systems to evaluate a variety of regimes and
environmental conditions. First, cases from the ENA XSAPR2 data will demonstrate the need for the new composite clutter map to improve the estimate of the baseline in environmental conditions with secondary sources of clutter that can be highly variable in time. Next, the CACTI radar CSAPR2 will demonstrate and cross compare the performance of using RHI scans as compared to PPI scans. Finally the XSACR and KaSACR data will show that the algorithm can scale to cloud radar frequencies, while also demonstrating some of the limitations that are unique to higher frequencies.



## 4.1 Composite clutter map evaluation

It is important that the generated clutter map only contain actual ground clutter. This can often be made more challenging in the presence of artifacts that can behave similar to ground clutter, such as sea clutter. Failing to estimate a good clutter map can severely affect the calculated RCA values for a radar. If for instance sea clutter points are selected in the map, the day-to-day

changes in sea state will be reflected in the 95th percentile of reflectivity over these points.

To demonstrate the importance of clutter map selection Figure 8a shows the clutter map from 13 March 2018 for the XSAPR2 radar at ENA. This day fits the requirements for a traditionally selected clutter map day (clear, no precipitation). However, sea clutter induced by winds across the surface of the ocean caused a false clutter signature. Figure 9a shows the resulting eRCA time series. The variability in the RCA values is high, with a mean value of over 7 dB. As per Wolff et al. (2015), this is outside

the +/- 1 dB value that would warrant adjustment. Figure 8b shows the results of applying the clutter map to a different day, 23 January 2018. A different clutter map on this day is caused by a difference in the ocean environment. There are, however, overlapping clutter points between both maps that are stable in time.

This problem can be mitigated by using the previously introduced composite clutter map technique. Figure 10 is the composite clutter map at ENA for XSAPR2 generated from the eRCA method. The steps outlined in Section 3.2 are followed to derive

Figure 10. Pixels in black (> 0.8) are designated as composite clutter points and are used to derive clutter area reflectivity. Pixels in gray (< 0.8) denote points that occurred less than 80% of the time in all daily clutter maps used in the composite. The black clutter points align with clutter point locations that are shared in both of the sample daily clutter maps from 13 March (Figure 8a) and 23 January (Figure 8b). This shows that the clutter points between 1-3 km range are stable and persistent clutter signatures. Note the gray, non-clutter points in the northern arc between 4-5 km range encapsulating the extent of clutter points

that appear in the same location in the 13 March clutter map (Figure 8a). These points do not make the cut off (>80%) for "stable clutter" and are not used to calculate clutter area reflectivity, whereas those same elements *are* used in the 13 March RCA calculation if the original algorithm is applied. These are likely the clutter points responsible for the large and fluctuating RCA values in Figure 9a.

Using the composite clutter map for ENA XSAPR2 (Figure 10) and 25 January 2018 for the baseline, Figure 9b shows daily

mean RCA values. These values fall well within an acceptable operational range of (+/- 0.5 dB), which is expected as the system was monitored by engineering staff during the period. A few daily RCA values toward the end of the observing period (23-27 March 2018) lie beyond the acceptable range, which indicates that there is an issue with radar calibration. It should be noted this indication does not diagnose the calibration issue, but rather serves as a starting point for troubleshooting. In this case, the radar transmitter started failing before ultimately reaching a point of complete failure.

## 4.2 Comparison of PPI and RHI eRCA

While the RCA method has been previously demonstrated to work for PPIs at S- and C-band (Wolff et al., 2015; Louf et al., 2019), applying it to RHI scans with less low elevation data presents a different set of challenges. The regular scan strategy



for the cloud radars during the CACTI field campaign did not include PPIs; preferring instead to maximize the time spent scanning vertical variability and relying on the presence of the C-band radar for horizontal spatial context. The application of the eRCA technique then depends on working with RHI scans if it is to be applied to the cloud radars for this campaign. The CSAPR2 C-band radar co-located at the campaign included both PPI and RHI scans allowing for cross-validation to compare

the performance of eRCA applied to both scan types and characterize the differences in performance before applying it to the RHI-only radars.

Figure 11 is a time series of daily RCA values calculated using PPI (black) and RHI (red) scans from CSAPR2 over a four month period during CACTI. Wolff et al. (2015) considers +/- 0.5 dB normal variability in reflectivity and defines a significant jump in RCA to be any value outside of the +/- 0.5 dB range. The median of both RHI- and PPI-based RCA values track the

zero line (during the time period the radar was monitored by engineering staff and did not experience any calibration drifts). This then lets us calibrate the cross-variability of these two techniques.

Histograms of RCA values from both PPI and RHI scans are shown in Figure 12. Both show a mean value under 0.5, with greater variability in the RHIs. This is expected owing to the smaller number of clutter points and clutter area reflectivity gates that can be used in the RHI calculation. In this case, the clutter map for PPI used 368% more gates than the RHI clutter map.

The standard deviation of the PPI-based estimate is 0.176 while the standard deviation of the RHI estimate is 0.238. When interpreting the RCA value from RHI scans, one should keep this increase in variability in mind.

One strong limitation of the eRCA technique for RHI scans is that the angles chosen for the RHI must include ground clutter. Similarly, the extent of scanning for the radar needs to include sufficiently low elevations to measure the surrounding ground clutter. For instance, of the twelve azimuths included in the CSAPR2 scan strategy, only 6 of these included points chosen

for the ground clutter map. Similarly there needs to be enough points in the RHI scans included as clutter to ensure robust statistics.

Even though the same reflectivity threshold and clutter map development method are used, the clutter map points of the CSAPR2 PPI clutter map and the CSAPR2 RHI clutter map are not necessarily the same points in space due to the differences in scanning. In addition to sample size differences, this difference in points can also account for much of the small-scale

variability in the RCA values in Figures 11-12

## 4.3   eRCA method for X-band radars

While we previously showed that the eRCA technique could be run on an X-band radar (XSAPR2 at ARM's ENA site), the unique setup at the CACTI site allows for a comparison between the effectiveness of the eRCA technique and direct reflectivity cross calibration. A comparison of reflectivity between systems is one of the most commonly used techniques to cross calibrate

co-located radar systems in field campaigns.

The XSACR ran only RHI type scans for the majority of the CACTI field campaign. As such, the RHI form of the eRCA technique was run on the XSACR radar for the duration of the CACTI campaign and is shown in Figure 13. The results show





that the RCA switches between two values but otherwise remains stable. This switch was found to be an error in the operational configuration of the radar that caused the wrong radar constant to be used for several time periods. The difference of 4.8 dB in the configuration change directly corresponds to the measured change on the RCA graph. There is some variability in the graph, especially later in April, that coincides with repeated transmitter issues from the XSACR. Due to a change in the range
gate spacing starting on 8 March 2019, a new composite clutter map and baseline calculated on 22 March 2019 were used for the time period beginning 8 March 2019 and the rest of the campaign.

To quantify the uncertainty in the RCA tracking, the difference in RCA values between CSAPR2 and XSACR were calculated and compared to the difference of reflectivity values for matched scans. The reflectivity cross comparison data was filtered to remove cases for which strong attenuation was expected. For the comparison, reflectivity was subtracted between the two
radars using the hemispherical RHI scans along the 6 scans that occurred every 30 degrees as shown in Section 2.2.1. Details of this filtering can be found in Appendix 1. Figure 14 shows that the two are highly correlated with an $R^2$ value of 0.965. This indicates that RCA for this case tracks the changing calibration between the two radars about as well as a direct reflectivity cross comparison, which is generally the gold standard of cross comparisons. Figure 19 (a) additionally shows the distribution of RCA values after correcting for the changing radar constant value as measured by RCA. This forms the distribution of the
residual and provides a standard deviation of 0.94 after correction by eRCA.

This characterization is important because the X-band is used to characterize the application of eRCA to millimeter-wave radars in the next section.

## 4.4  eRCA method for Ka-band radars

The wide deployment of Ka-band radars with ARM facilities (where every scanning SACR cloud radar has a Ka-band compo-
nent) makes the automatic tracking of calibration for millimeter-wave radars a desirable goal. Millimeter-wave radars however can be very sensitive to small perturbations in the environment and operating conditions making tracking of calibration difficult. The application of the eRCA technique to the KaSACR radar was ultimately successful, but processing required additional care and pre-processing steps as detailed here and Appendix 1.

The first run of eRCA on the KaSACR data from CACTI resulted in very high variability during many time periods despite
the strict attenuation filtering detailed in Appendix 1. When periods with rain falling at the radar are removed to limit radome attenuation, much of the high variability persisted. The output of the individual file level (sub-daily) RCA values shown in Figure 15 for one 72 hour case highlights this issue. During time periods when there is rain on site, radome attenuation causes a large change in RCA values, which is a known limitation of higher frequency radars. However, even during time periods when there is no measured precipitation on site, RCA values track the relative humidity. This is the case for many days with large
humidity changes (not shown). The change in RCA values with humidity, on occasion surpassing 10-15 dB, is too large to be explained by gaseous attenuation induced by propagation between the radar and clutter points through the humid environment. Even at $25°C$ the change from 0% to 100% humidity only results in a few dB of two-way attenuation at the ranges of clutter used by the eRCA method. It is believed that radome attenuation, caused by condensation in the humid environment, is the



culprit. The hydrophobic coating on the radome has not been replaced within the last 10 years, leading to a failure in its ability to shed water.

Interestingly, the comparison hints at the ability for sub-daily corrections using eRCA. While these sub-daily corrections are not fully explored in this paper, the variability in RCA estimates at shorter time scales appear small enough to correct at a time

resolution of less than one day. This behavior was verified for several different three day periods during the campaign.

Rather than sub-daily corrections due to environmental factors, the goal of this work is to correct systematic biases in radar calibration. Based on this analysis, a relative humidity filter was implemented in the processing for the KaSACR. Figure 16 (a) shows the entire file level RCA output for the campaign colored by relative humidity above 90%. It is clear that most of the high RCA variability is caused by high humidity times. Based on this finding, a file level filter for humidity above 90% was

implemented. This is in addition to the propagation attenuation filtering that was implemented to remove time periods where attenuation was expected between the radar and the clutter targets as documented in Appendix 1. The RCA result after filtering is shown in Figure 16 (b) with greatly reduced variability.

After filtering the data for attenuation and high relative humidity, the sub-daily measurements are combined into a daily measurement in Figure 17 along with two fitted curves. There is an increasing trend in RCA which corresponds to a decrease

in transmit power that was noted by radar engineering. On 18 March 2018, the waveguide was dismantled and a blockage was cleared out. This blockage, likely there since installation, caused a decrease in sensitivity, and as such, a new curve was fitted to the data to account for the jump discontinuity upon its removal. The continued degradation of the transmitter following removal can still be observed however.

To both validate and cross-compare the eRCA technique applied at Ka-band, the KaSACR data was cross compared with

the co-aligned XSACR radar. Figure 18 shows the comparison of the difference of reflectivities compared to the difference in RCA. As before, if RCA is accurately tracking the relative calibration, then these two parameters will match within some constant offset. While the slope (0.845) and the $R^2$ show that RCA is indeed tracking the XSACR, there is larger variability than for the comparison of X- and C-band in Figure 14. This is not entirely unexpected as the humidity and attenuation filters previously implemented are not perfect. Additionally, we are not explicitly controlling for gaseous attenuation, which provides

another source of uncertainty.

The KaSACR initially had a large calibration offset of 6-10 dB as calculated from measurements using a corner reflector and comparisons with self consistency from the CSAPR2, and if the calibration of a radar drifts too far, it becomes necessary to do an initial calibration correction pass to get the radar back within the ballpark of correct calibration. For example, if reflectivities are 10+ dB lower than expected, then clutter bins fail to make it past the initial clutter selection procedure. A

similar behavior affects the direct reflectivity cross comparisons where cloud types with too much attenuation are chosen for the cross comparison. Further details of the filters and comparisons can be found in Appendix A **??**

The distribution of KaSACR RCA values shows a much greater variability than the C- or X-band. This can be seen in Figure 19 that cross-compares X- and Ka-band histograms of RCA values. This can be caused by greater variability in clutter power for the Ka-band due to its higher frequency, but is primarily caused by the dropping power of the transmitter in time. If we





compare the RCA of the residual after correction by the polynomial fit in Figure 17, we get a standard deviation of 1.24. This is the variation in clutter power after correction and can be inferred as a metric for the uncertainty of the eRCA technique for the KaSACR.

Despite greater variability, eRCA can be used to correct systematic changes in the radar calibration at Ka-band. Indeed,
eRCA shows potential to correct more transient issues in the calibration.

## 5   Summary and future work

This study proposes an extension of the RCA technique (eRCA) for use with range-height scans and higher frequency radars. The eRCA method was successfully applied to X-band PPI scans at the ENA ARM observatory. Increased variability in the clutter field at the ENA site prompted an extension of the clutter map development method, which utilizes only the most stable
clutter points from a selection of observed days to establish a composite clutter map. The necessity of a composite clutter map in a region with a variable clutter field is proven with ENA XSAPR2 data. The proposed eRCA clutter map selection method mitigates these situations.

The eRCA technique was successfully extended for use with RHI scans, shown with a comparison of C-band PPI and RHI data from the CACTI field campaign. To accommodate RHI scans, the range of the clutter map is extended out to 40 km and
the lowest 5 degrees of elevation angles are considered. RCA values derived from CSAPR2 PPI and RHI scans were within an acceptable margin of variability (+/- 1.0 dB) that matches engineering records for the radar.

Results from a cross-comparison of CSAPR2, XSACR, and KaSACR at the ARM site during CACTI validated use of the eRCA technique for X-band RHIs. Through a cross-comparison of the co-mounted X- and Ka-band radars, the eRCA technique was shown to be valid for Ka-band radars as well as long as additional constraints were applied. The variability in the Ka-band
RCA is greater than for X-band and is more sensitive to environmental and operational parameters of the radar. For the unique situation at the CACTI site, attenuation and high relative humidity filters drastically improved the performance of the eRCA technique.

The eRCA technique is not intended for real-time calibration adjustments, but it is useful for a-posteriori calibration. The technique is shown to capture changes in calibration due to radar performance, as as well engineering changes. For this reason,
the eRCA technique can be used as a tool for monitoring the health of the radar. Correcting the data when changes in calibration exceed +/- 1.0 dB occur is crucial for accurate quantitative precipitation estimation and model evaluation.

Future work with the eRCA technique needs to address the role of changing meteorological conditions and suitability of locale. Most current applications have been either tropical or oceanic where the seasonality of ground clutter is limited, or relatively short duration that fails to fully capture seasonal variations in the environment. At many mid- and high-latitude loca-
tions, the ground clutter signature can change seasonally. For example, snow in the Arctic changes the surface backscattering properties. This necessitates a methodology for determining when to update a clutter map to cope with changing conditions.



Similarly, RCA has not to our knowledge been evaluated in environments where the presence of evolving snow and ice conditions can modulate the radar cross section of environmental targets. Finally, this work suggests that eRCA may have the potential for monitoring and correcting radome attenuation, which deserved further investigation.

*Code and data availability.* The original data used in this study can all be found in the DOE ARM archive (https://www.arm.gov/data).

The code developed in this work is available and can be found on Github at https://github.com/josephhardinee/rca with an archival copy located on figshare at time of release.

## Appendix A: Filters and Processing

### A1 Attenuation Filters

#### A1.1 Path-integrated attenuation filter

Higher frequency radars are subject to larger path attenuation. As such this attenuation needs to be accounted for before using eRCA so that path attenuation does not bias the results of using clutter on days with precipitation. To mitigate the effect of this data was filtered using a stringent path-integrated attenuation filter. For each radar ray path-integrated attenuation is calculated and all rays that exceed a set threshold are removed from the 95th percentile calculation. The equation used for specific attenuation with relation to reflectivity is:

$$A_H = aZ_H^b \tag{A1}$$

where a and b are coefficients that vary based on radar band, and attenuation is given in dB km$^{-1}$. The coefficients for the three radar bands used in this study are provided in Table 1.

Path-integrated attenuation is calculated along each ray as:

$$IA_H = 2\int_0^r A_H(r)dr \tag{A2}$$

A threshold of 0.1 deg km$^{-1}$ was set, where rays that exceeded the threshold were not used in calculations. This filter is designed to be quite stringent and serves to reduce the biasing and variability of eRCA.

### A2 Reflectivity Cross Comparisons

#### A2.1 Gate-matching

The co-mounted and co-located radars at the CACTI site allowed for direct comparison of reflectivities in space and time.

The scan strategies for the radars were synchronized to provide as much temporal overlap as possible. Given the varying gate-





**Table 1.** Attenuation coefficients used for path-integrated attenuation filtering, where $A_H = aZ_H^b$

| Band | a | b |
|------|------|------|
| Ka | 0.00115481 | 0.95361079 |
| X | 0.000372 | 0.72 |
| C | – | – |

C-band attenuation not calculated

spacing and beamwidths of the three radars (see Table 1 in main text), the individual radar gates must be matched to each other in order to directly compare observations. In these comparisons, the radar with the smaller beamwidth and gate spacing is matched to the nearest azimuth or elevation and range of the radar with the larger beamwidth.

## A2.2    Rejection criterion

5    In order to compare reflectivities between radars, we select for light precipitation conditions, focusing on low reflectivities (below 25 dBZ) and meteorological correlation coefficients (greater than 0.95). Table 2 includes threshold values used for the X-band vs. Ka-band comparison and the X-band vs. C-band comparison. Path-integrated attenuation filtering (identical to the method and thresholds used in Table 1) was also included to perform the comparisons.

After discovering a correlation between relative humidity near the surface and Ka-band RCA, a relative humidity filter was
10    applied to remove RCA data points during periods of relative humidity greater than 90%. Since this filter removes all relative humidity values greater than 90%, this filter also serves to remove time periods when there was rain on site over the radar.



**Table 2.** Thresholds for weather and data selection used in radar reflectivity cross-comparisons.

| Band Comparison | X - Ka | X - C |
|---|---|---|
| Reflectivity range ($Z_H$, dBZ) | -5 - 15 | -5 - 25 |
| Correlation coefficient minimum ($\rho_{HV}$) | 0.95 | 0.95 |
| Path-integrated attenuation maximum ($IA_H$, deg km$^{-1}$) | 0.1 | 0.1 |
| Elevation range (deg) | 10 - 170 | 10 - 170 |

*Author contributions.* AH: Implementation and analysis of eRCA code, writing of paper. JCH: Evaluation and guidance of the project, writing of paper. NB: Evaluation of results. AV: Evaluation of results and guidance on COR environment. AM: Cross comparisons with KAZR radar and additional processing.

*Competing interests.* The authors declare that they have no conflict of interest.

5  *Acknowledgements.* This research was primarily supported by the Office of Biological and Environmental Research of the US Department of Energy as part of the Atmospheric Radiation Measurement (ARM) Climate Research Facility, an Office of Science scientific user facility. Dr. Varble's contributions were supported under the U.S. Department of Energy Office of Science Biological and Environmental Research as part of the Atmospheric Systems Research Program. PNNL is operated for DOE by Battelle Memorial Institute under contract DE-AC05-76RL01830.

10  The authors would like to thank Peter Argay, Vagner Castro, Tercio Silva, Juarez Viegas, Bruno Cunha, Todd Houchens, and Andrei Lindenmaier for their work deploying and maintaining the radars during the CACTI field campaign that provided much of the data used in this study.





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

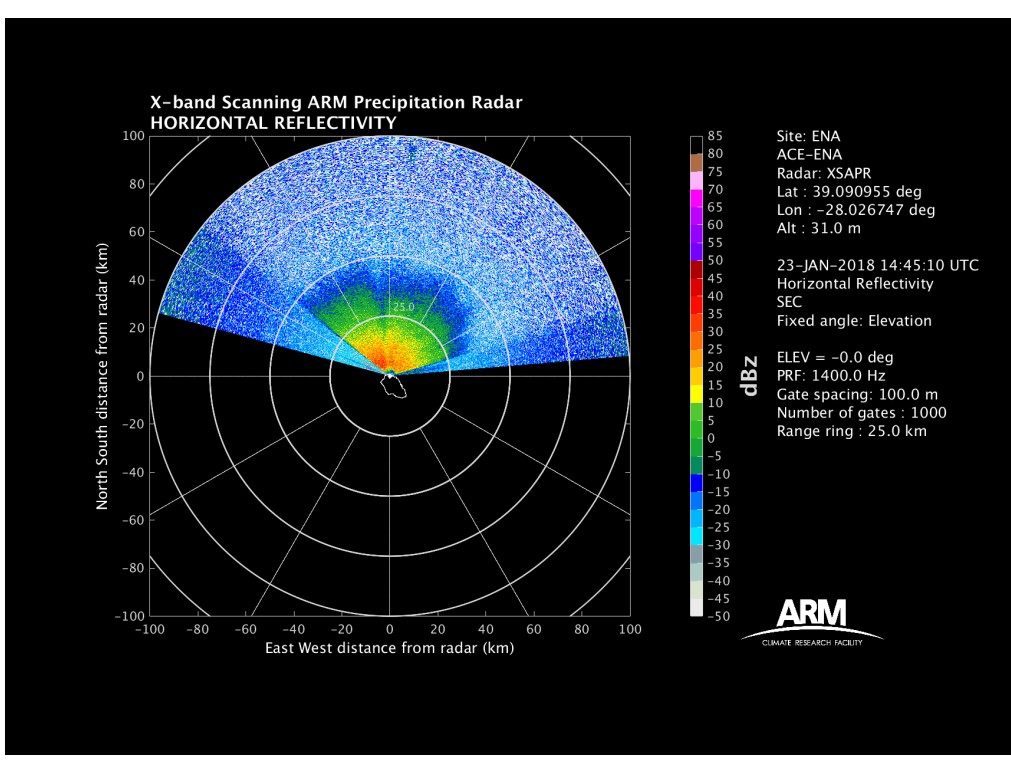

**Figure 1.** The Eastern North Atlantic XSAPR2 X-band PPI from 23 January 2018 14:45:10 UTC. The radar returns in the northern sector near the radar are sea clutter, not precipitation. The sea clutter varies day to day with wind conditions presenting a challenge for the traditional RCA algorithm. The southern sector is not scanned, due to mountain blockage as shown by the outline of the island.

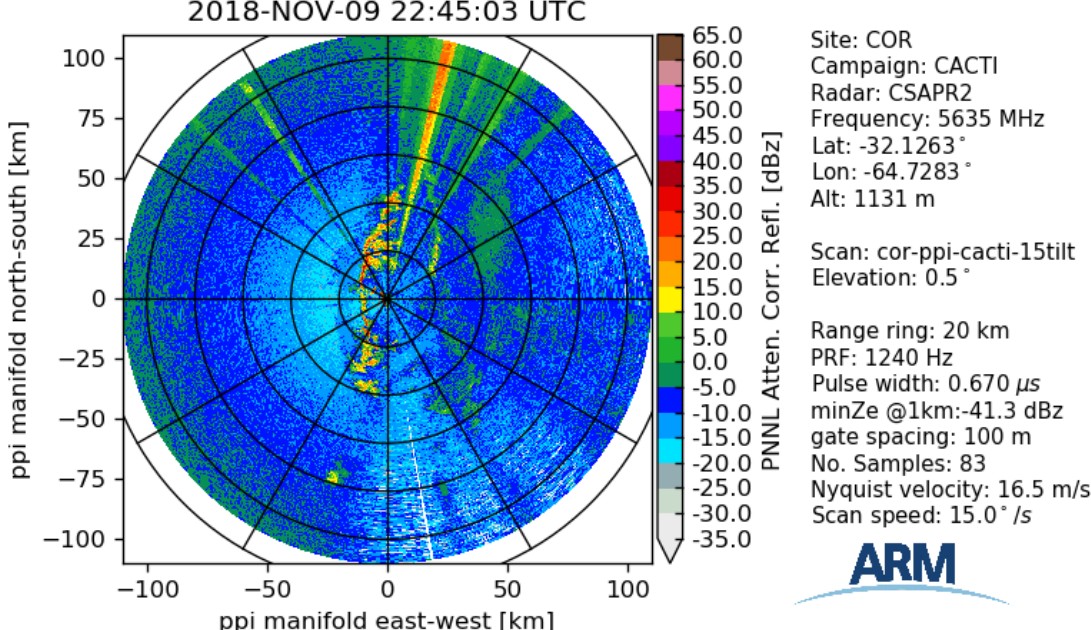

**Figure 2.** The CACTI CSAPR2 C-band radar PPI from 9 November 2018 22:45:03 UTC. The Sierras de Córdoba lie west of the radar, visible in the ground clutter signature. There is also significant clutter due to the hills to the east of the radar. There are also 3 interference sources visible to the northeast and the northwest. The northwest source is attributable to a transmission tower, while the northeastern is the town of Córdoba.



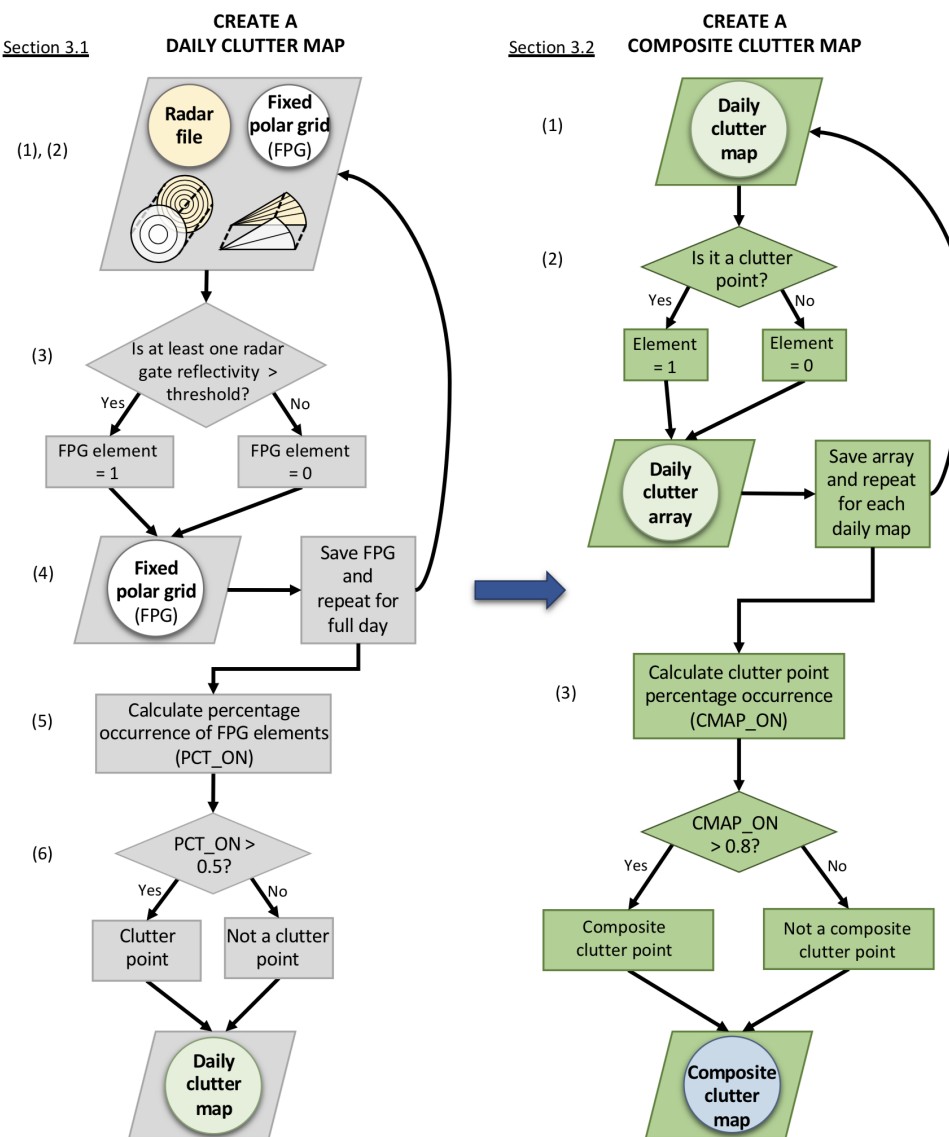

**Figure 3.** Workflow for designing the clutter map used to calculate the baseline value. The first chart shows how a clutter map is created from a single day of data. The second chart shows how multiple daily maps can be combined into a composite map used in this work that is more resistent to temporal fluctuations.



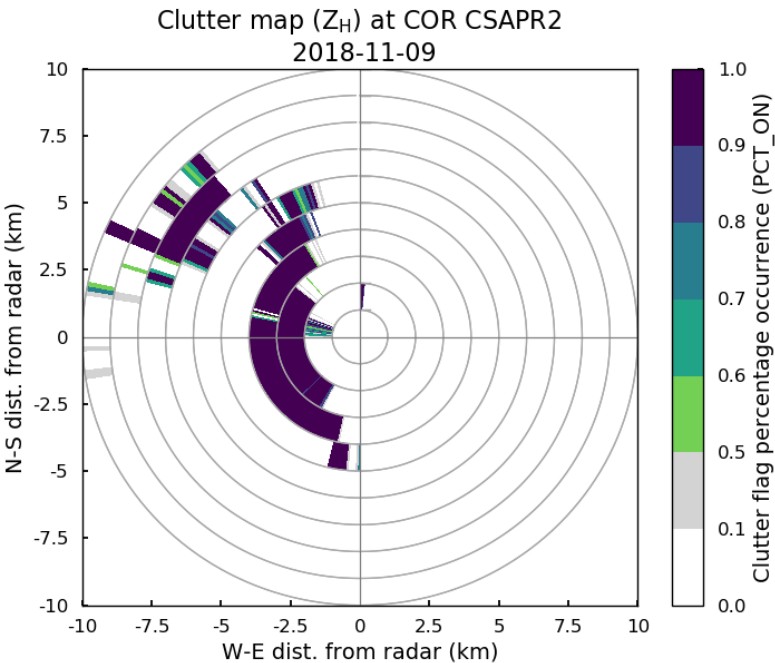

**Figure 4.** The clutter map calculated from the CACTI CSAPR2 PPI scans on 9 Novemebr 2018. The map predominantly uses the clutter caused by the hills to the west of the radar. The clutter to the east of the radar is out of range of the 10 km limit for eRCA.



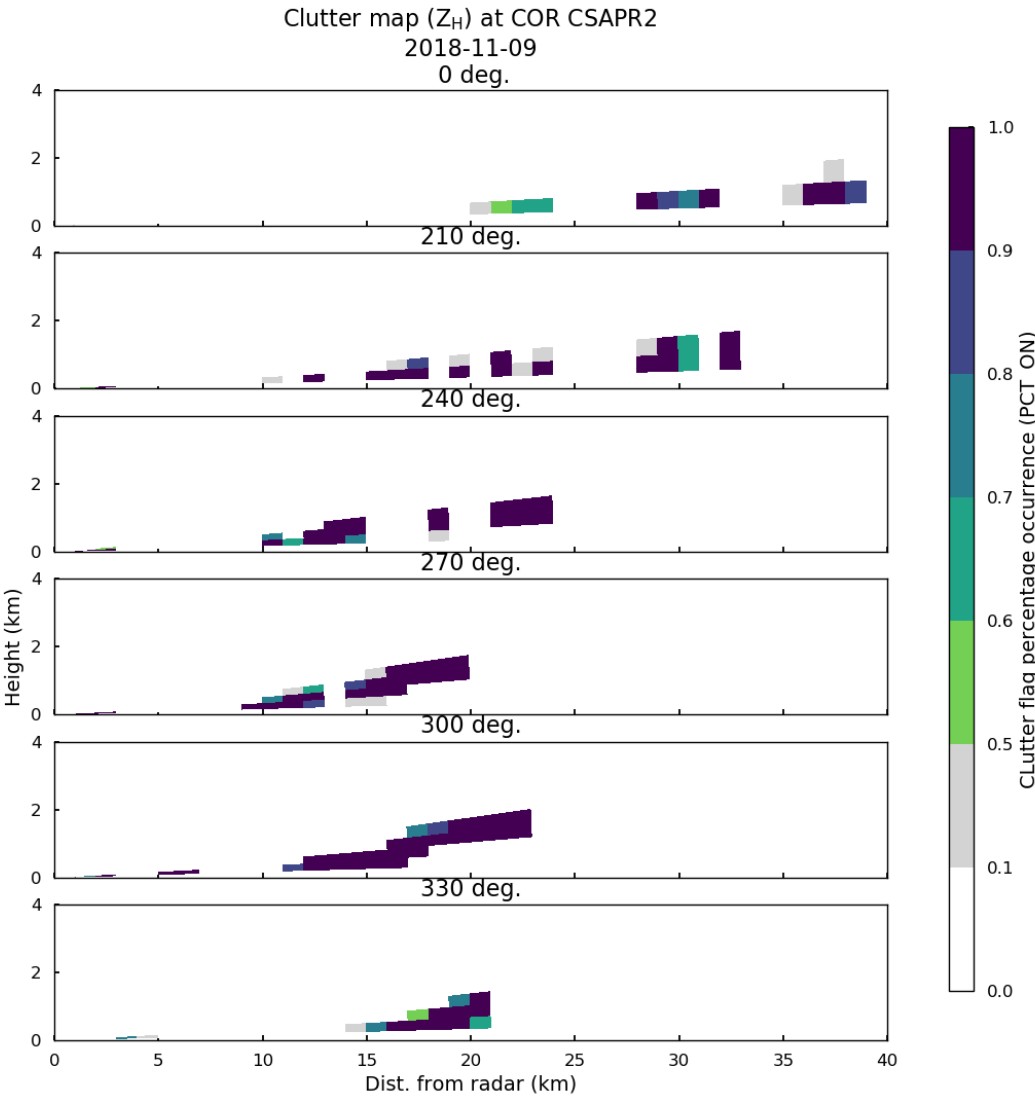

**Figure 5.** The clutter map calculated from the RHI scans from 9 November 2018 for the CACTI CSAPR2 radar by azimuth angle. Angles for the RHI were every 30 degrees, with full hemispherical scanning. Similar to the PPI map, the clutter used is to the west of the radar.





**Figure 6.** Workflow for calculating a baseline value from the composite clutter map and radar files. The second diagram shows how the baseline value can be used to calculate a daily adjustment to reflectivity.

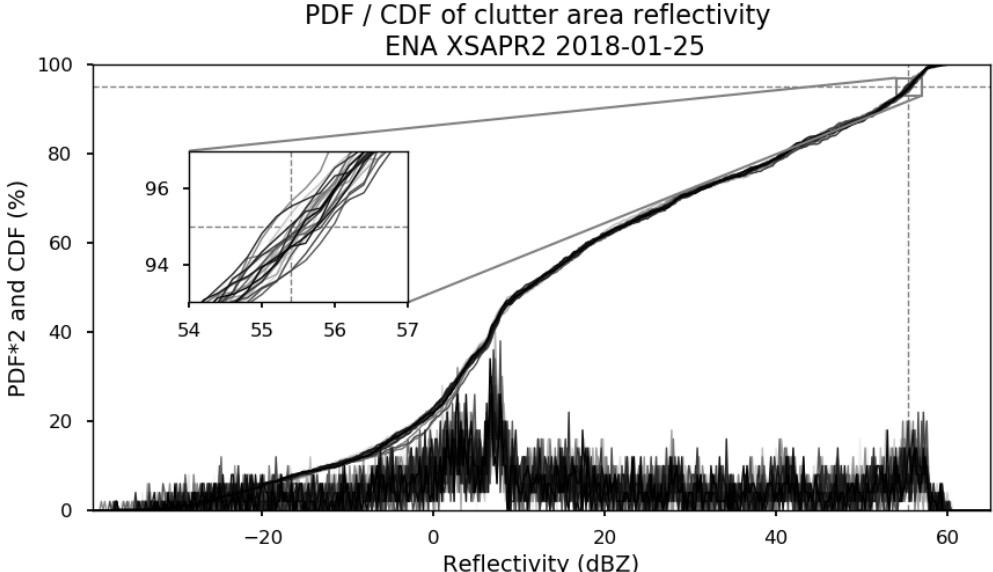

**Figure 7.** Probability density functions and cumulative distribution functions shown from each hour during the selected baseline day, 25 January 2018, at ENA XSAPR2. The 95th percentile is marked with the corresponding reflectivity value also marked at 55.4 dBZ. This is the value used for the dBZ95$_{baseline}$ in the RCA calculation. The inset shows the stability of the 95th percentile for use in the RCA value.



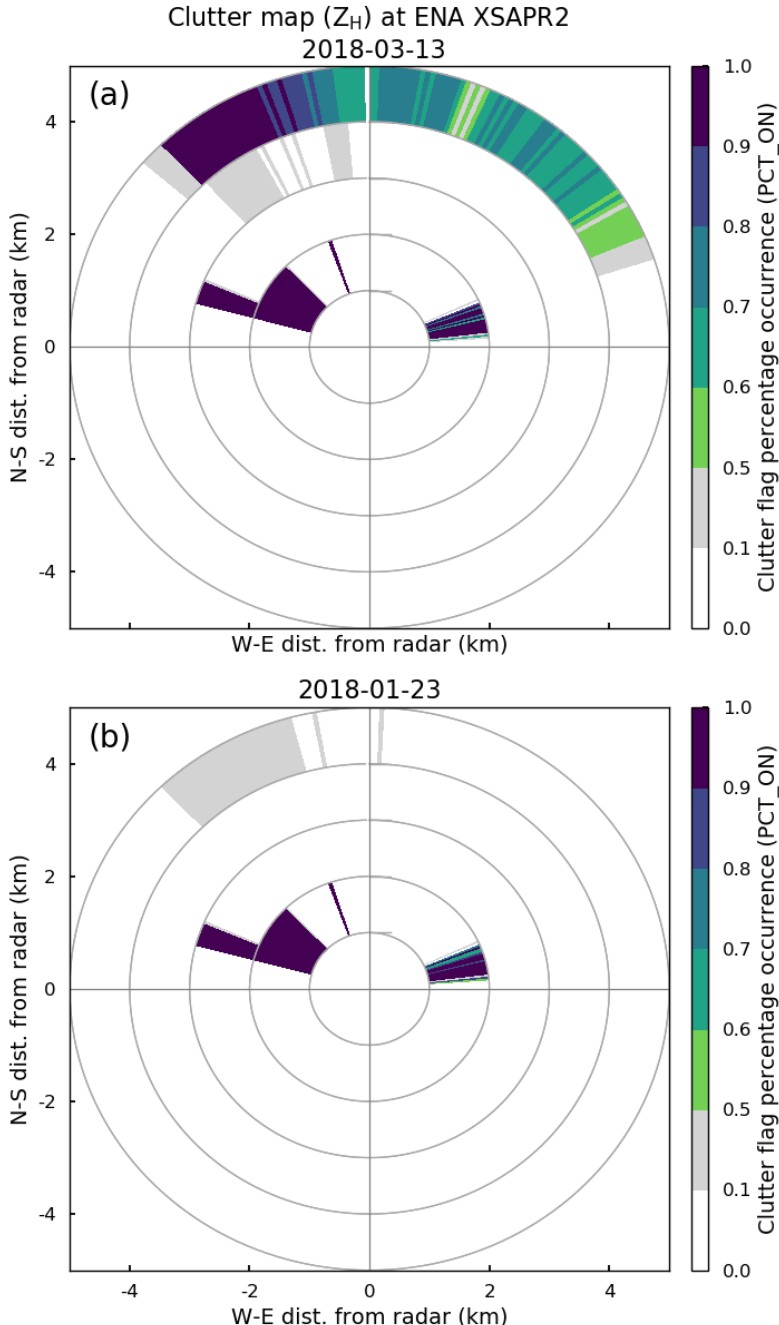

**Figure 8.** Daily clutter map for XSAPR2 PPIs at the ENA site for 13 March 2018 (top) and 23 January 2018 (bottom). These maps show how the variability of ground clutter (primarily induced by the significant sea clutter around the site) can cause the clutter choices to change. There are, however, points in common that are stable regardless of time period used as seen to the north within 3 km of the radar in both images.



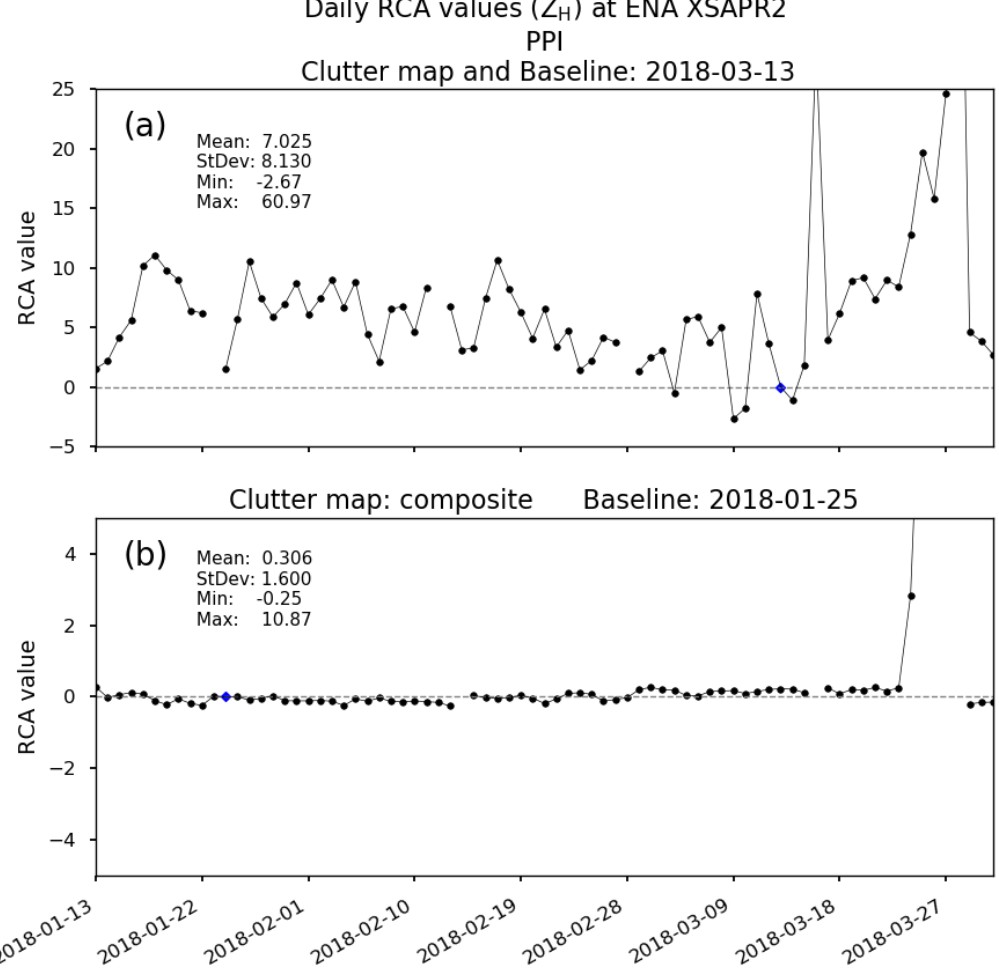

**Figure 9.** RCA time series for XSAPR2 at the ENA site calculated using 13 March 2018 for the clutter map and the baseline (top) and using the composite clutter map and 25 January 2018 for the baseline (bottom). The high variability in the RCA value is reduced by using a composite map that accounts for daily temporal stability of the clutter.

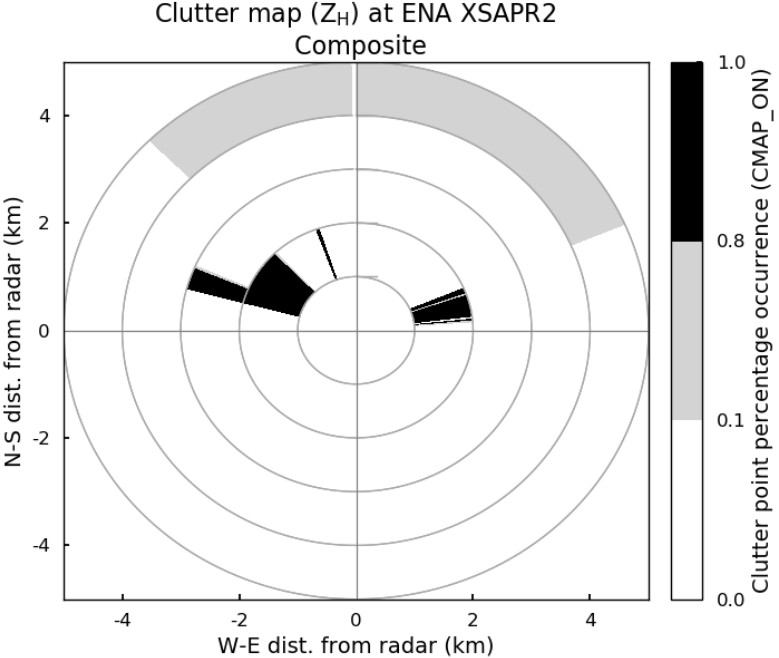

**Figure 10.** Composite clutter map for XSAPR2 PPIs at the ENA site, created using clutter points from every fifth observation day that occurred at least 80% of the time. The black denotes those time periods that were used for the composite map. The light gray denotes points that made it into a daily clutter map, but were not stable across multiple maps.



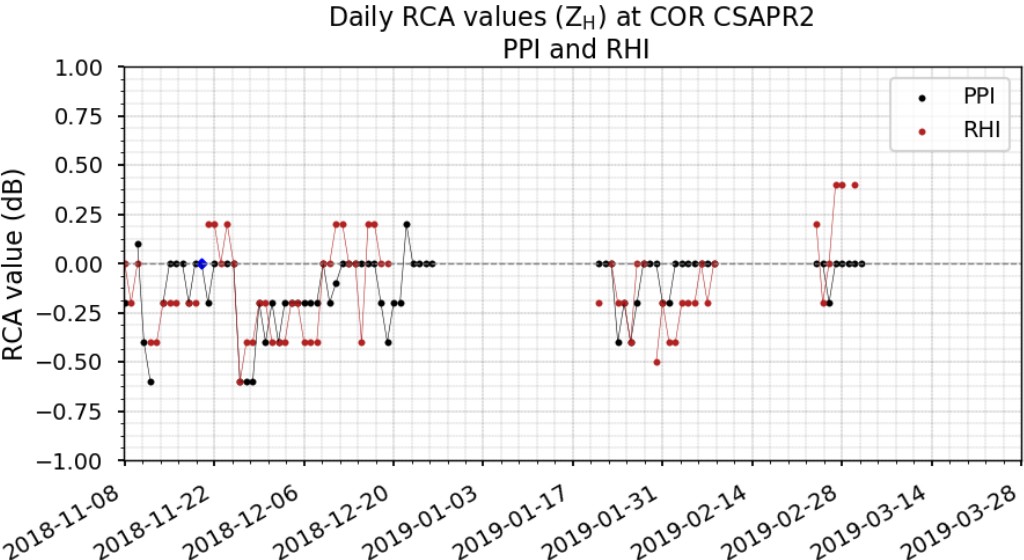

**Figure 11.** RCA time series for CSAPR2 PPIs and RHIs at the COR site during CACTI. RCA calculated using the composite clutter maps and 20 November 2018 as the baseline. The two radars are within the statistical noise of the eRCA technique. The gaps in the time series are due to periods where the radar was offline.



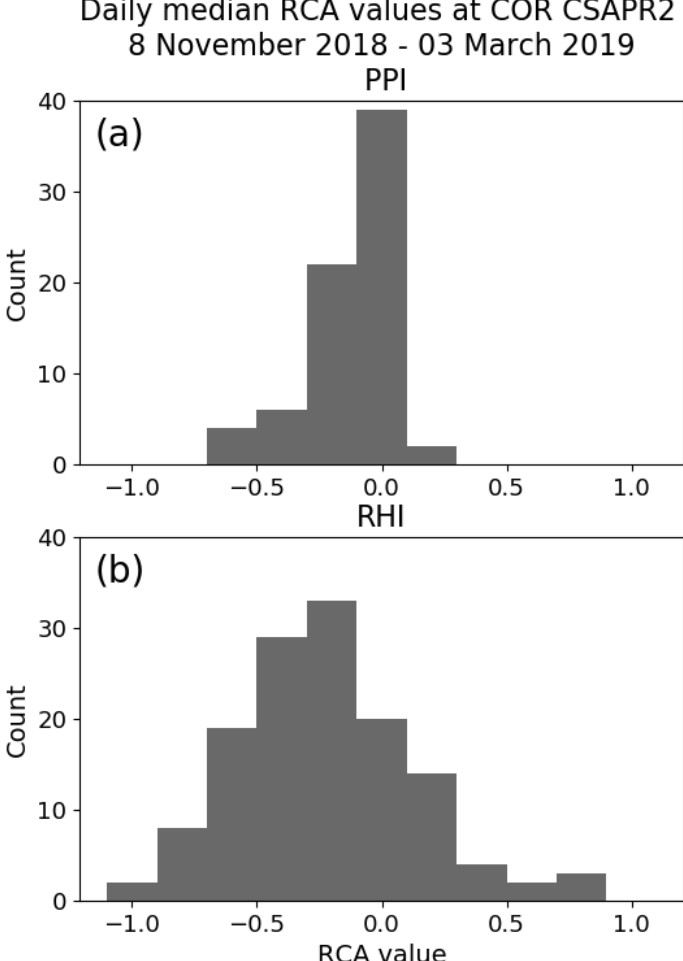

**Figure 12.** Histogram of daily mean RCA values for COR CSAPR2 between 8 November 2018 - 03 March 2019. The variability of RCA values is demonstrated to fall well within +/- 1 dB. Additionally the variability of the eRCA technique using the two different scan types is comparable.

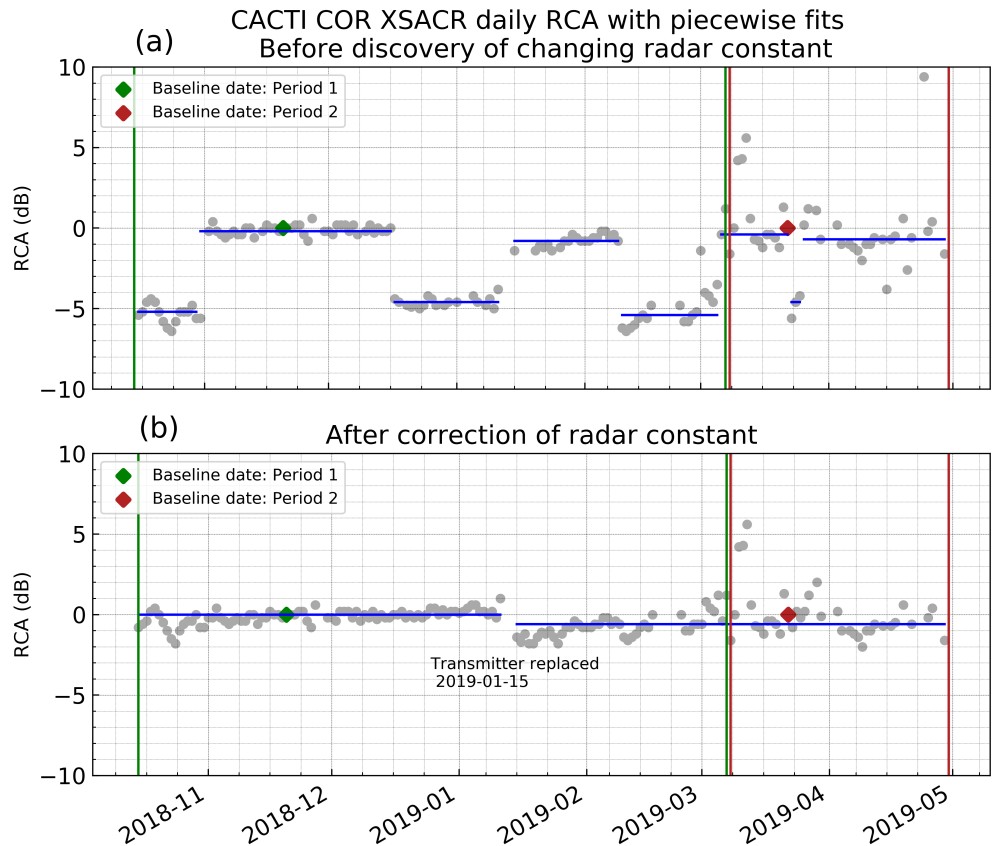

**Figure 13.** RCA time series for CACTI XSACR. (a) corresponds to the RCA run without any corrections. The values jump between two piece-wise constant values caused by changing radar constants. The vertical bar shows when the operational scan strategy was changed from 25m to 75m gates and so a new composite clutter map was calculated. The RCA is shown in (b) after correcting for the piecewise jumps.



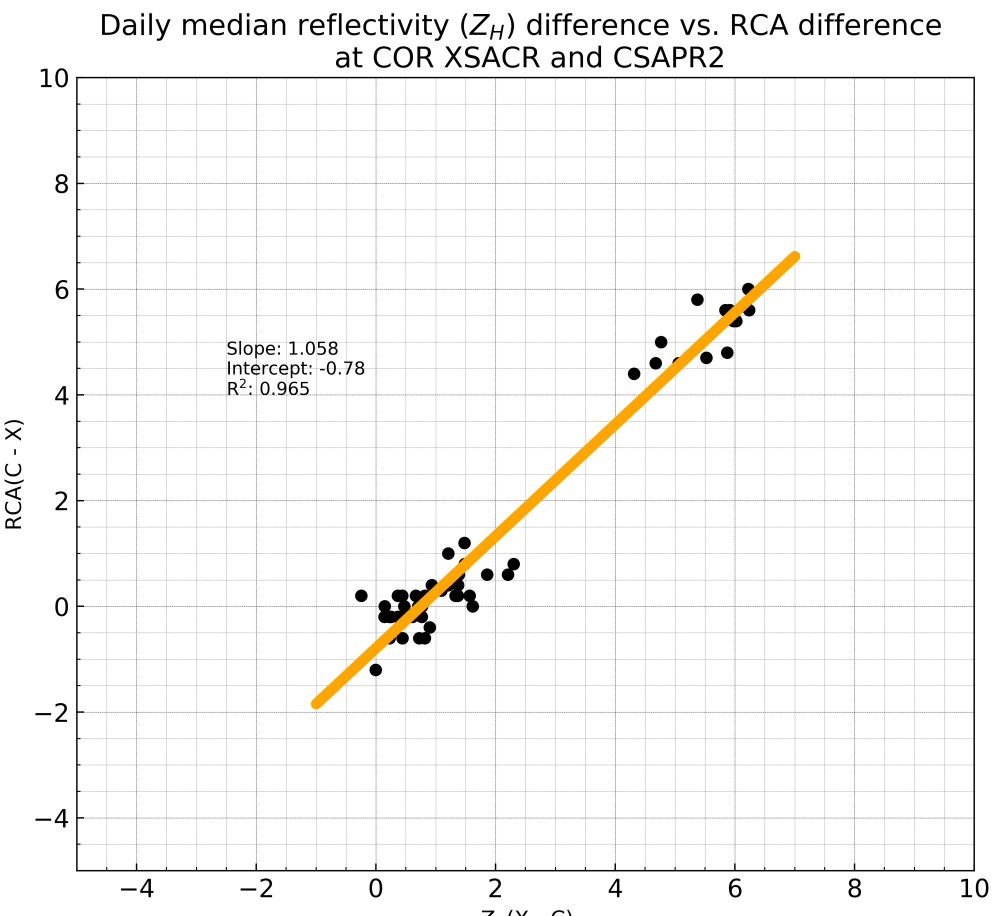

**Figure 14.** The scatter plot of the difference of RCA values between CSAPR2 and XSACR compared to the difference in mean filtered reflectivities between the two radars. This shows that RCA is capturing the change in reflectivity calibrations as well as the direct comparison.

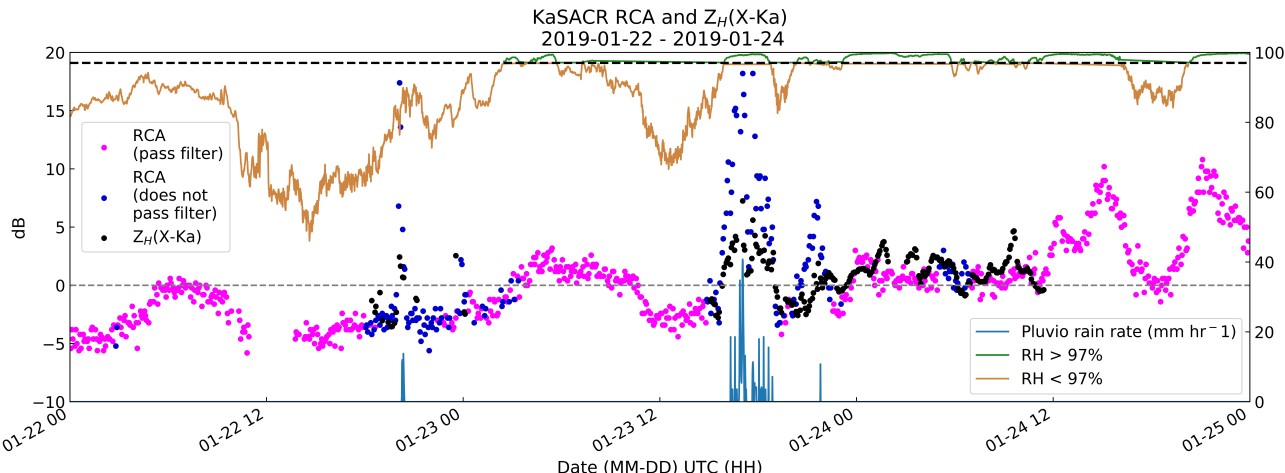

**Figure 15.** Sub-daily RCA for KaSACR at the COR site during a 72 hour period between 22 January 2018 and 25 January 2018. Sub-daily RCA is plotted by color, based on scan times that passed a relative humidity threshold of 97%. Rain rate from the Pluvio-2 rain gauge is shown to coincide with periods of high RCA values. Reflectivity cross comparison between XSACR and KaSACR is also plotted to compare and identify patterns suspected to be caused by changes in relative humidity or precipitation.



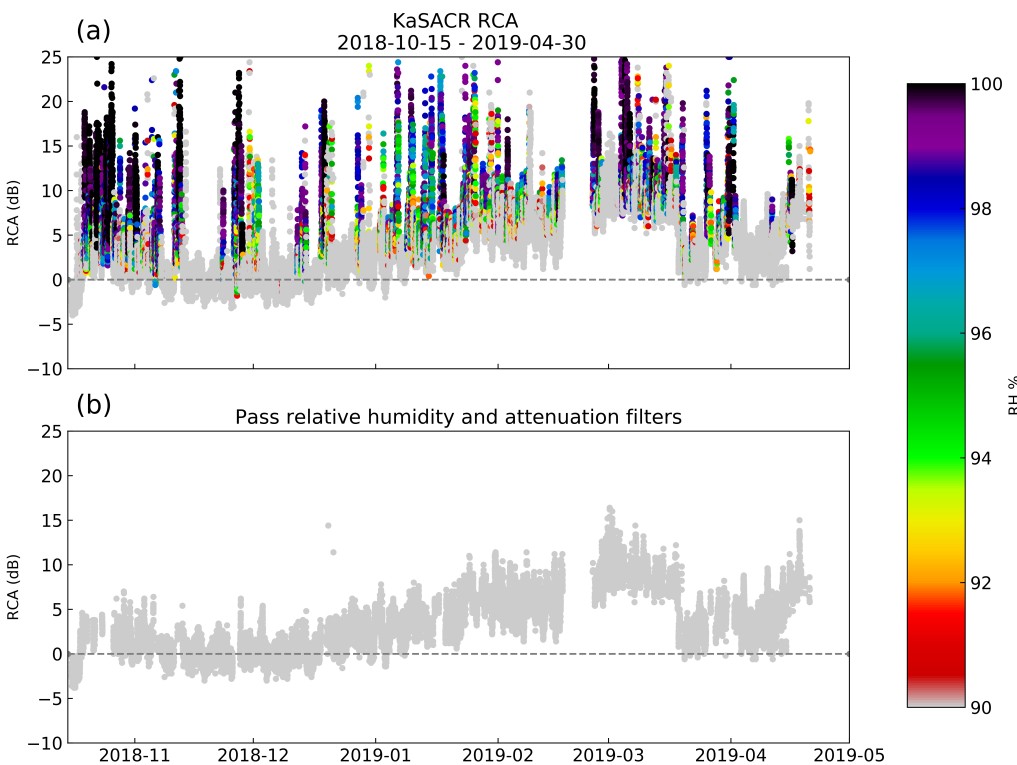

**Figure 16.** (a) shows the sub-daily (per file) RCA values without any pre-filtering for the CACTI field campaign. Color indicates relative humidity with grey as the threshold at 90%. Significant variability is present on days with higher relative humidity. The sub-daily values after filtering for attenuation and 90% relative humidity are shown in (b). Filtering reduces the variability of the data allowing for a clear trend to be observed.

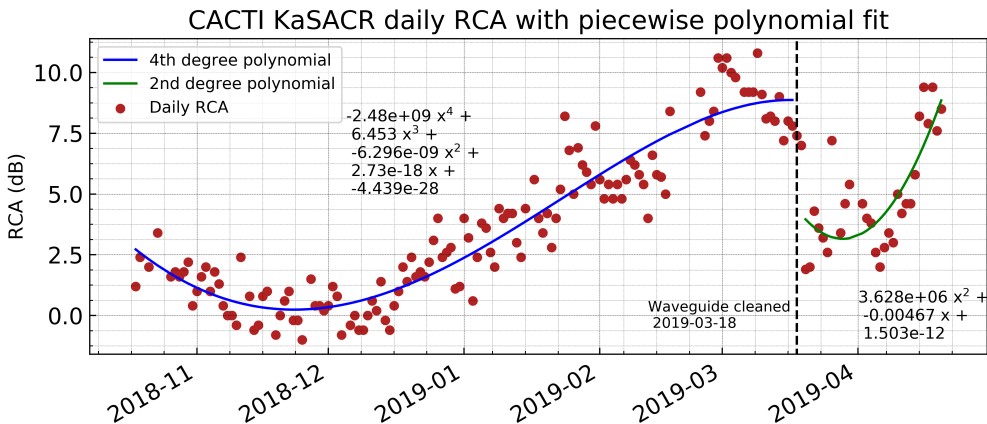

**Figure 17.** The RCA daily values for the CACTI KaSACR radar are shown with a corresponding piece-wise polynomial fit. A decreasing trend in power resulted in an increasing trend in the RCA value. On March 18th, 2019, the waveguide had an obstruction removed and a sharp decrease (due to corresponding increase in effective power) can be observed. As such a new polynomial fit is calculated for this time period. However, the general trend of decreasing power and increasing RCA continues. The fit is given relative to the epoch time in seconds of each day.

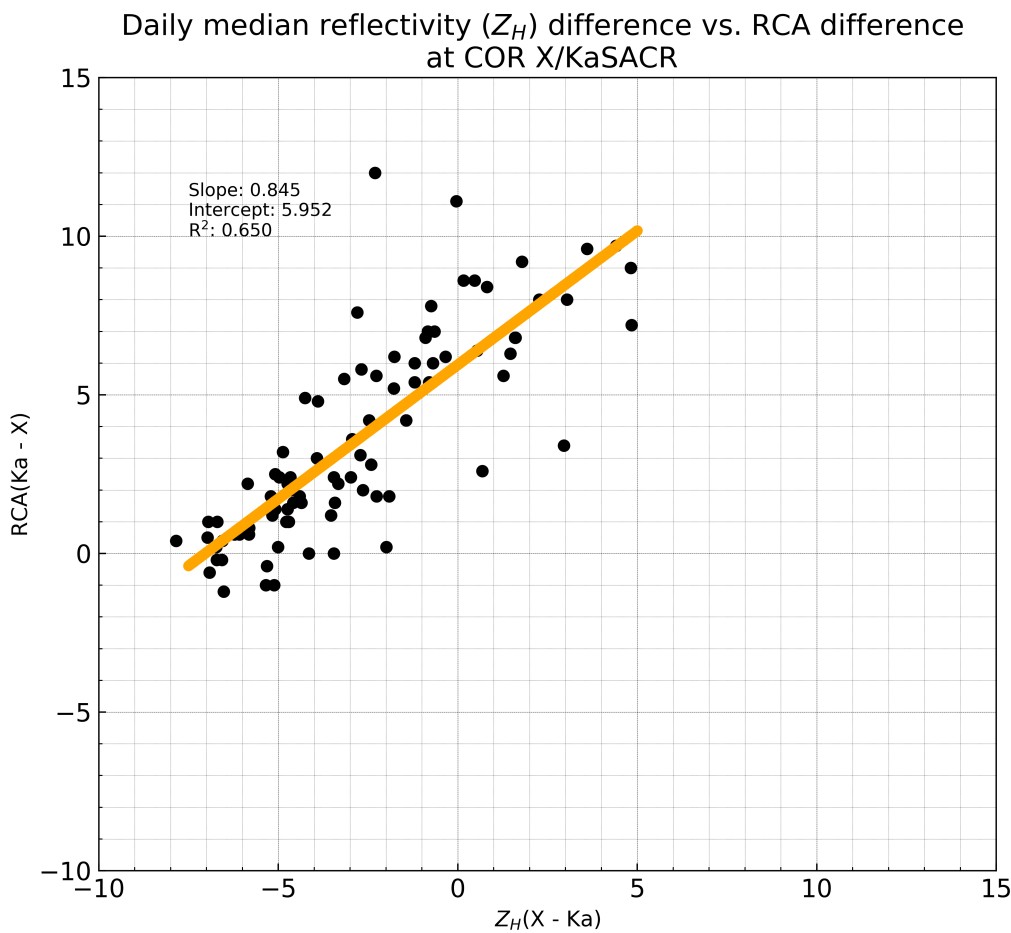

**Figure 18.** The scatter plot of the difference of RCA values between the Ka- and X-band radars as compared to the difference in mean filtered reflectivities for the two radars. This shows that RCA is capturing the change in reflectivity calibrations as well as the direct comparison.

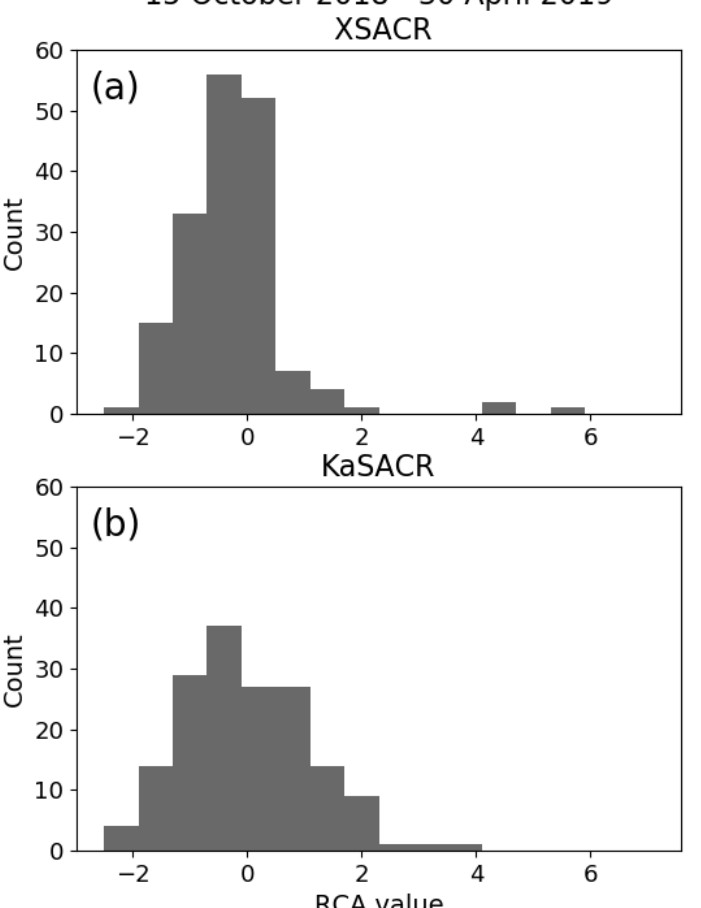

**Figure 19.** Histograms of daily mean RCA values for COR XSACR and KaSACR between 8 November 2018 and 30 April 2019. The XSACR RCA residual is shown in (a) after correction for piecewise jumps. The KaSACR residual after correction by the piecewise polynomial curve is shown in (b).



**Table 1.** Characteristics and specifications for ARM radars used in this study.

| Radar | CSAPR2 | XSAPR2 | XSACR | KASACR |
|---|---|---|---|---|
| Site | COR | ENA | COR | COR |
| Frequency | 5.7 GHz | 9.5 GHz | 9.71 GHz | 35.3 GHz |
| Wavelength | 5.26 cm | 3.15 cm | 3.09 cm | 8.50 mm |
| Transmit power | 350 kW | 350 kW | 20 kW | 2 kW |
| Antenna diameter | 4.3 m | 5.3 m | 1.82 m | 1.82 m |
| Beam width | 0.9 | 0.5 | 1.4 | 0.33 |
| Typical range resolution | 100 m | 100 m | 30-150 m | 30-150 m |
| Polarization | Dual | Dual | Dual | Transmit horizontal linear |
| Scan Types | PPI | PPI | RHI | RHI |
|  | RHI | RHI |  |  |



**Table 2.** Thresholds and ranges for different radar frequencies for generation of the composite clutter map.

| Band | Reflectivity threshold (dBZ) | Range limit for PPI (km) | Range limit for RHI (km) |
|------|------------------------------|--------------------------|--------------------------|
| S | 55 | 1 - 10 | N/A |
| C | 45 | 1 - 10 | 1 - 40 |
| X | 35 | 1 - 5 * | 1 - 40 |
| Ka | 10 | N/A** | 1 - 20 |

*Exclusive to XSAPR2 at the ENA site. For a typical X-band PPI, range may be 1-10 km **PPI-based eRCA not calculated for KaSACR