# Peer review of "An Extended Radar Relative Calibration Adjustment (eRCA) Technique for Higher Frequency Radars and RHI Scans."

_Atmospheric Measurement Techniques, 2020_

## Referee Comment (RC1) · Anonymous Referee #1 · 27 Mar 2020

Authors: Alexis Hunzinger, Joseph C. Hardin, Nitin Bharadwaj, Adam Varble, and Alyssa Matthews

This paper extends the Relative Calibration Adjustment (RCA) technique originally developed by Silberstein et al. 2008 for an S-band radar at a tropical oceanic site at Kwajalein, Republic of the Marshall Islands. Wolff et al. 2015 showed the method also worked well over land at several mid-latitude sites using NASAs S-band NPOL radar. Louf et al. 2019 used the RCA technique and NASA satellite-based radar retrievals to correct a nearly 20-year period of C-band CPOL observations in Darwin, Australia

The primary advancements that this paper provides are: 1) extension to using RHI

as well as PPI data; 2) additional robustness of the clutter map by combining multiple maps over different days to produce a composite map that is more representative of true clutter (and thus not so strongly affected by more transient returns from sea-spray et al.); and, 3) adaptation of the technique to higher frequencies, including X-band and Ka-band. The paper is well written and concise and contributes significantly to the radar community in that it allows for a posterioi calibrations to radar reflectivity data, as well as near-real-time health monitoring of radar systems.

The eRCA was first applied to the CSAPR2 (C-band) radar that was recently deployed during the CACTI field campaign in Argentina. CSAPR2 was able to perform both PPI and hemispherical PPIs over several angles. Conventionally, the authors use the 0.5 elevation scan in the PPIs to calculate the RCA, but also develop a novel approach to the RCA by using the lowest 5 of the RHIs to calculate an independent RCA estimate.

This paper does provide a substantial contribution to scientific progress in that it extends a very powerful method to monitor the relative calibration of radars at higher frequencies than have been used. They also improve the robustness of the method by proposing a composite clutter map whereby they use multiple days for determining the clutter area reflectivity. They also extend the original PPI-based method to RHIs, which is an important improvement for radars that do not routinely perform PPI scans.

The discussion in the Appendices is very useful, particularly Appendix 1 that discusses PIA filtering necessary for utilizing the RCA at higher frequencies. Overall this is an exceptional paper that should be published.

Minor comments:

1) Table 1 in Appendix 1 should probably be labelled Table A1.1 to prevent being confused with Table 1 in the main text. Similarly so for Table 1 in Appendix 2 to be labelled Table 2A.1

2) Appendix 1: Can you please provide some more detail on how the co-mounted and

co-located radars are configured? Are the independently scanning? Maybe a picture would help.

---

## Referee Comment (RC2) · Anonymous Referee #2 · 4 Apr 2020

This manuscript proposes an extended version of the relative calibration adjustment (eRCA) technique for weather radar applications. In particular, the extension and applications are focused on range-height scans and higher frequency radars (C to Ka band). The eRCA method was demonstrated using DOE-ARM radar measurements from different field sites.

Overall, this manuscript is very well written, and it is easy to follow. I enjoyed reading this work. I recommend publication of this manuscript.

One quick comment about the FPG resolution of 1-km × 1-deg: Given the possible challenge to collect sufficient clutter data from a statistical point of view, would it be

more effective to increase the FPG resolution, especially at Ka-band which has much higher resolution measurements?

A minor edit: On page 6 (line 25), please remove extra "the".

---

## Author Comment (AC1) · 5 May 2020

Response to Reviewer 1

We would like to thank the reviewer for their time reading and reviewing our paper and their feedback which will improve the paper.

"The paper is well written and concise and contributes significantly to the radar community in that it allows for a posterioi calibrations to radar reflectivity data, as well as near-real-time health monitoring of radar systems. The eRCA was first applied to the CSAPR2 (C-band) radar that was recently deployed during the CACTI field campaign

in Argentina. CSAPR2 was able to perform both PPI and hemispherical PPIs over several angles. Conventionally, the authors use the 0.5 elevation scan in the PPIs to calculate the RCA, but also develop a novel approach to the RCA by using the lowest 5 of the RHIs to calculate an independent RCA estimate. This paper does provide a substantial contribution to scientific progress in that it extends a very powerful method to monitor the relative calibration of radars at higher frequencies than have been used. They also improve the robustness of the method by proposing a composite clutter map whereby they use multiple days for determining the clutter area reflectivity. They also extend the original PPI-based method to RHIs, which is an important improvement for radars that do not routinely perform PPI scans. The discussion in the Appendices is very useful, particularly Appendix 1 that discusses PIA filtering necessary for utilizing the RCA at higher frequencies. Overall this is an exceptional paper that should be published."

We thank the reviewer for their very kind words. In regards to the appendix, it is something we felt very strongly should be included in more radar processing papers, where the details of the processing is laid out and we are glad that sentiment is shared.

"Minor comments: 1) Table 1 in Appendix 1 should probably be labelled Table A1.1 to prevent being confused with Table 1 in the main text. Similarly so for Table 1 in Appendix 2 to be labelled Table 2A.1 "

The reviewer is absolutely correct and it was our mistake in the LaTex file that caused incorrect numbering. We thank you for catching this. The document has been updated and Figures are now labeled A1, A2 in accordance with AMT style.

"2) Appendix 1: Can you please provide some more detail on how the co-mounted and co-located radars are configured? Are the independently scanning? Maybe a picture would help."

The radars each have their own antenna co-mounted on the same pedestal and are aligned prior to each field campaign. We agree an image of the radars would help the

readers to understand their unique layout. We have included a figure showing the Ka/X SACR deployed at the CACTI field campaign in Argentina as figure 2.

Again we would like to thank the reviewer for their time and feedback.

---

## Author Comment (AC2) · 5 May 2020

Response to reviewer 2

We would like to thank the reviewer for their time reviewing this manuscript, their kind words, and the feedback that will surely make this a better paper.

This manuscript proposes an extended version of the relative calibration adjustment (eRCA) technique for weather radar applications. In particular, the extension and applications are focused on range-height scans and higher frequency radars (C to Ka band). The eRCA method was demonstrated using DOE-ARM radar measurements

from different field sites. Overall, this manuscript is very well written, and it is easy to follow. I enjoyed reading this work. I recommend publication of this manuscript. We would like to thank the reviewer for their kind words.

"One quick comment about the FPG resolution of 1-km $\times$ 1-deg: Given the possible challenge to collect sufficient clutter data from a statistical point of view, would it be more effective to increase the FPG resolution, especially at Ka-band which has much higher resolution measurements?" We thank the reviewer for the comment and it provides us an opportunity to clear up a potential misunderstanding in the text. The FPG itself is primarily treated as a mask to detect areas of clutter. Then the gates within FPG elements identified as clutter are all used individually. In this way we can hopefully reject more clutter in the initial choice of masks and keep to clutter signatures that are much more stable, while still retaining richer statistics when actually calculating the RCA CDF. To make this more clear we have added the following sentence on page 8, Line 27 While the FPG clutter map is on a fixed coarsened grid, the PDF/CDF are calculated using all of the range gates within each FPG element that was determined to be clutter.

"A minor edit: On page 6 (line 25), please remove extra "the" " This has been corrected.

Again we would like to thank the reviewer for their kind comments and positive feedback.